# Exploring the Longitudinal Stability of Food Neophilia and Dietary Quality and Their Prospective Relationship in Older Adults: A Cross-Lagged Panel Analysis

**DOI:** 10.3390/nu15051248

**Published:** 2023-03-01

**Authors:** Hanna R. Wortmann, Ulrike A. Gisch, Manuela M. Bergmann, Petra Warschburger

**Affiliations:** 1NutriAct-Competence Cluster Nutrition Research, Berlin-Potsdam, Germany; 2Department of Psychology, Counseling Psychology, University of Potsdam, Karl-Liebknecht-Str. 24-25, 14476 Potsdam, Germany; 3German Institute of Human Nutrition Potsdam-Rehbruecke, Arthur-Schneunert-Allee 114-116, 14558 Nuthetal, Germany

**Keywords:** food neophilia, dietary quality, NutriAct Family Study, cross-lagged panel analysis, healthy eating

## Abstract

Poor dietary quality is a major cause of morbidity, making the promotion of healthy eating a societal priority. Older adults are a critical target group for promoting healthy eating to enable healthy aging. One factor suggested to promote healthy eating is the willingness to try unfamiliar foods, referred to as food neophilia. This two-wave longitudinal study explored the stability of food neophilia and dietary quality and their prospective relationship over three years, analyzing self-reported data from *N* = 960 older adults (*M*_T1_ = 63.4, range = 50–84) participating in the NutriAct Family Study (NFS) in a cross-lagged panel design. Dietary quality was rated using the NutriAct diet score, based on the current evidence for chronic disease prevention. Food neophilia was measured using the Variety Seeking Tendency Scale. The analyses revealed high a longitudinal stability of both constructs and a small positive cross-sectional correlation between them. Food neophilia had no prospective effect on dietary quality, whereas a very small positive prospective effect of dietary quality on food neophilia was found. Our findings give initial insights into the positive relation of food neophilia and a health-promoting diet in aging and underscore the need for more in-depth research, e.g., on the constructs’ developmental trajectories and potential critical windows of opportunity for promoting food neophilia.

## 1. Introduction

Eating a varied, balanced, and healthy diet throughout life helps protect against malnutrition in all its forms, as well as against a variety of diet-related non-communicable diseases, including type 2 diabetes mellitus, cardiovascular diseases, and cancer [1,2]. Although the advantages of a healthy diet are clear, many individuals worldwide do not adhere to dietary guidelines [3]. The fact that poor dietary quality is a leading cause of morbidity [4] makes the promotion of healthy dietary patterns even more of a priority to reduce non-communicable diseases [3].

Given the significant increase in life expectancy worldwide [5] and the increasing risk of chronic diseases and multimorbidity with age [6], older adults are a particularly important target group for promoting healthy eating. A systematic review underlines that nutrition is a key determinant of chronic disease in later life, highlighting the importance of a favorable diet for healthy aging, in terms of both physical and mental health, and thus for quality of life in older age [7]. In addition, the period of older age is characterized not only by significant normative life changes (e.g., the transition from work to retirement [8]), but often by other major life events (e.g., marital transitions [9], widowhood [10], or changing health conditions [11]), all of which can lead to a disruption of long-standing eating routines. Hence, this period may represent an important window of opportunity for dietary change [8,12], highlighting its potential for intervention strategies to promote healthy eating, and thus to enable healthy aging. 

To account for the complex nature of diet, the study of overall dietary patterns has emerged in recent decades as a promising alternative to conventional approaches that focus on single nutrients or foods [13]. Dietary patterns can be developed either exploratory or based on predefined patterns constructed from evidence regarding nutritional health, such as dietary guidelines. The latter a priori approaches allow for the calculation of diet quality indices that assess the diet overall, such as the well-investigated Mediterranean Diet (MD) Score [14]. Although dietary patterns are increasingly investigated in nutritional epidemiology, little is known about their stability over time, especially in older age [15]. In fact, only a few studies have examined prospective changes in dietary patterns in older adults. Whereas Samieri et al. [16] found no change in the adherence to the MD over 13 years in a large sample of older women, Hill et al. [17] found evidence that the overall dietary quality of older women declined over 14 years. Thorpe et al. [15] observed the stability of two exploratory dietary patterns in a sample of older adults over four years, in which the overall dietary quality increased, but only in men. Using a latent class analysis, Harrington et al. [18] found high dietary stability among older adults over a ten-year period. Overall, the current evidence is based on different methodological approaches and samples and appears to be inconsistent but indicates a certain stability of dietary patterns at older ages. Noticeably, all of the analyzed dietary patterns were either exploratory or based solely on dietary guidelines. Knowledge of the longitudinal stability of dietary quality based on indices that additionally include current evidence for chronic disease prevention [19] could lead to a more profound understanding of dietary intervention opportunities aimed at reducing the burden of chronic disease in later life. 

To develop effective intervention programs to promote health-beneficial diets in older adults, it is essential to understand not only the stability of dietary quality, but also the correlates and determinants of healthy eating. The multitude of food decisions people make daily is undoubtedly based on a complex interplay of various factors [20]. One of the many factors suggested to influence food choices is the willingness to try unfamiliar foods [21]. Described as an evolutionarily beneficial survival mechanism, individuals are inherently ambivalent towards unfamiliar foods that can provide not only a new and possibly nutritious food source, but also the risk of consuming something potentially harmful or poisonous. Faced with this conflict, known as the omnivore’s dilemma [22], individuals differ greatly in their food neophilic and neophobic tendencies [23]. While food neophilia manifests in the overt willingness to try new and unfamiliar foods [24], food neophobia describes the avoidance of and reluctance to eat novel foods [25]. 

Previous research has focused primarily on food neophobia [26], which was found to be negatively associated with dietary variety [27] and dietary quality, e.g., measured cross-sectionally by adherence to the MD [28] and key aspects of Nordic dietary recommendations [29] and prospectively by adherence to the Baltic Sea Diet Score over an eight-year period [30]. Moreover, cross-sectional studies have shown that food neophobia is related to a lower fruit and vegetable intake [27,31] and a reduced willingness to try healthful food alternatives [32]. In addition, food neophobia was shown to be associated with different health-related biomarkers and an increased risk of type 2 diabetes mellitus [30], as well as an increased BMI [29]. Overall, the previous evidence on food neophobia in adults underscores its role as a barrier to a health-promoting diet [27] and its potential health risks [30]. 

It has long been assumed that food neophilia and food neophobia were merely opposite poles of the same continuum between approaching and avoiding unfamiliar foods [33], which has resulted in little research explicitly addressing food neophilia. However, recent evidence on the distinction between food neophobia and neophilia suggested a separate consideration of food neophilia in future research, as the two constructs appear to be closely related but conceptually distinct [34,35]. Focusing on positive food-related attitudes and preferences (such as food neophilia) rather than maladaptive ones also corresponds to the idea of positive psychology [36] and has appeared to be a promising approach to promoting a healthy diet in recent years [37,38]. Studies on food neophilia in the context of nutritional research are very scarce and, to our knowledge, exclusively cross-sectional. With both studies using brief screening instruments to measure dietary quality, Lavelle et al. [39] found a small positive association between food neophilia and dietary quality, whereas McGowan et al. [40] found no association between food neophilia and a brief measure to distinguish between healthy and unhealthy food choices, but did find a small association with a lower intake of saturated fat, which is another aspect of dietary quality. 

Thus, while there is cross-sectional evidence that food neophilia and diet quality are positively related, no study has yet examined the longitudinal relationship between food neophilia and dietary quality. Although positive cross-sectional relations indicate that food neophilia might play a beneficial role in healthy dietary behavior, prospective evidence is needed to verify the assumption that food neophilia precedes healthy eating. Due to the cross-sectional nature of the previous studies, alternative theoretical assumptions concerning the direction of the effect between food neophilia and dietary quality are also possible. For example, it is conceivable that not only food neophilia has a positive effect on dietary quality, but also vice versa, e.g., that a higher dietary quality is associated with better cooking and food skills [39] and greater food knowledge [41], which in turn may lead to greater interest in and willingness to try unfamiliar foods. Studies addressing the longitudinal reciprocal relationship between food neophilia and dietary quality will help elucidate their interrelation over time, potentially providing important implications for intervention programs to promote a health-beneficial diet.

Focusing on older adults and the possibilities for promoting healthy aging, knowledge of food neophilia in this age group and its stability across time will further expand our understanding of intervention opportunities. However, there is only limited research on food neophilia in older individuals. When comparing different age groups in non-representative samples in terms of their mean levels of food neophilia, Wortmann et al. [35] found no significant differences, while Van Trijp [42] found significantly higher levels of food neophilia in younger adults than in older adults. In a prospective study of young adults leaving their parental homes, Meiselman et al. [43] found a high stability of food neophilia during this particular period of change. To date, however, it remains unclear whether food neophilia is comparably stable during the period of older age, and whether food neophilia tends to decrease or increase in later life. 

To overcome the described gaps in the literature, the main purpose of the present study was to examine the stability of both food neophilia and dietary quality in older age (defined here as ages 50 and older), as well as their prospective reciprocal relationship, with a higher dietary quality referring to a higher adherence to dietary recommendations for the prevention of chronic diseases. Analyzing the longitudinal data from older adults participating in the Nutritional Intervention for Healthy Aging (NutriAct) Family Study (NFS), in a cross-lagged panel design, we addressed the following research questions: How stable are food neophilia and dietary quality in older adults over a three-year period? Does food neophilia predict dietary quality over time and vice versa? Based on the previous evidence, we hypothesized a high stability of food neophilia and dietary quality over time. Given the limited evidence on the association between food neophilia and dietary quality, their prospective reciprocal relationship was examined in an exploratory manner.

## 2. Materials and Methods

### 2.1. Study Design and Procedure

Data collection for the first (T1) and second (T2) waves of the NFS took place between January 2017 and March 2019, and between September 2020 and November 2021, respectively. The NFS is a web-based, prospective, interdisciplinary study examining food choices from psychological, epidemiological, and sociological perspectives and is part of the NutriAct competence cluster funded by the German Federal Ministry of Education and Research. Study participants were recruited in groups of two or more family members (spouses and siblings). The recruitment procedures are described in detail elsewhere [44]. In brief, the recruitment of the families was based on an index person who had already participated in the European Prospective Investigation into Cancer and Nutrition (EPIC)-Potsdam study. Before inclusion in the study, all participants were asked to provide written informed consent. After enrollment in the study, the participants received login credentials for their personalized web-based questionnaires. To participate in the second wave (T2), those participants who had completed the T1 questionnaires were invited by mail. For both waves, the online questionnaires each consisted of four coherent parts [44]. They included a comprehensive dietary intake survey, as well as a set of reliable and valid instruments, to assess the potential factors influencing food choices based on the DONE framework [20]. 

### 2.2. Participants

For the present study, participants aged 50 years and older were included, resulting in a final sample of *N* = 960 participants from 409 families who completed the T1 online questionnaires. Of these participants, *N* = 829 from 372 families took part at T2, resulting in a dropout rate of 13.6%. The mean interval between T1 and T2 was 40.0 months (*SD* = 4.2 months). An overview of the sample characteristics at T1 and T2 is presented in Table 1. 

### 2.3. Measures

#### 2.3.1. Food Neophilia

Food neophilia was measured with the German version [35] of the Variety Seeking Tendency Scale (VARSEEK) [46]. The VARSEEK consists of 8 items (e.g., “I am curious about food products I am not familiar with.”), that were rated on a 7-point Likert scale, ranging between 1 (*completely disagree*) and 7 (*completely agree*), i.e., higher mean values reflect higher food neophilia. Previous research has supported the scale’s internal consistency (Cronbach’s α = .93) and its test-retest reliability (*r* = .87) [35]. Positive correlations with ratings of willingness to try unfamiliar foods and openness as well as negative correlations with food neophobia and general neophobia support the scale’s construct validity [35]. For the present study, Cronbach’s α as well as McDonald’s ω [47] were .93 at T1 and T2.

#### 2.3.2. Dietary Quality

Dietary quality was calculated using a multi-step approach, resulting in the NutriAct diet score [19], a new diet score based on the current evidence for chronic disease prevention and the guidelines of the German Nutrition Society. In the first step, participants’ usual food intake was assessed following Knüppel et al. [48]. For this purpose, the food intake probability was calculated based on the repeated application of four 24 h food lists (24 h-FL) [49] over 12 months, enriched by frequency information from an EPIC-Potsdam Food Frequency Questionnaire-II (FFQ2) [50]. The food intake probability was then multiplied by a person-specific daily consumption amount derived from a reference population from the representative German National Nutrition Survey II (NVS II) [51], resulting in an estimate of the participants’ usual intake of a variety of different food items. In the second step, the evidence-based scoring scheme of the NutriAct diet score was applied, as described in detail by Jannasch et al. [19]. For this purpose, food items were first aggregated into a total of 10 food groups that were shown to be associated with the most common non-communicable chronic diseases, such as type 2 diabetes mellitus, cancer, and cardiovascular diseases. Depending on the food group, either the daily or weekly portions of the food groups were then calculated for each participant, based on the participants’ usual intake of the respective food items. To calculate the final NutriAct diet score, the intake per food group was rated up to one point, resulting in a NutriAct diet score of between 0 and 10 points. For example, given the health-promoting effect of fruit consumption, the higher the intake category (ranging between never and 2 or more portions/week), the higher the respective points (0, 0.5, 1). In summary, a higher NutriAct diet score indicates a higher dietary quality according to the current evidence for chronic disease prevention and the recommendations of the German Nutrition Society [52].

#### 2.3.3. Additional Database of Government Policies during the COVID-19 Pandemic

As the T2 data were collected during the COVID-19 pandemic, we used the stringency index retrieved from the Oxford COVID-19 Government Response Tracker (OxCGRT), a global panel database capturing government policies related to containment, health, and economic policies during the COVID-19 pandemic [53], to quantify and statistically control for the stringency of government containment and closure policies in Germany at each survey date. The stringency index varies with time and can range between 0 (*no measures*) and 100 (*total lockdown*). In the present study, the mean T2 stringency index was *M* = 64.65 (*SD* = 11.98, range = 42.59–85.19).

### 2.4. Statistical Analyses

First, descriptive analyses were performed. Pearson correlation coefficients were calculated to investigate the bivariate relationships between food neophilia and dietary quality at T1 and T2 (*r* > .10 small, *r* > .30 medium, *r* > .50 large effect sizes; [54]). Holm-Bonferroni corrections were performed to adjust the level of significance for multiple testing [55]. Wald tests were performed for both the study variables to compare their respective mean values at the time points T1 and T2. To estimate the effect sizes, Cohen *d* (*d* = 0.2 small, *d* = 0.5 medium, *d* = 0.8 large effect size) [56] was calculated [57]. 

As the participants were nested within families, the preliminary analyses included an examination of the intraclass correlation coefficients (ICC) as a quantitative measure of the similarity among the observations within families. The ICCs were computed by employing the type “twolevel basic” option in Mplus. Moreover, a dropout analysis was performed to determine whether the study variables were systematically associated with dropout after T1, using independent samples *t*-tests for the interval variables (age, weight status, food neophilia, dietary quality) and chi-square tests for the categorical variables (gender, educational status). Our dropout analysis revealed no significant differences between the participants who dropped out after T1 and those who remained in the study in terms of gender, χ^2^ (2) = 0.16, *p* = .924; age, *t*(162) = 0.72, *p* = 0.471; weight status, t(958) = −1.35, *p* = .179; educational status, χ^2^ (2) = 2.42, *p* = .299; food neophilia, t(958) = −0.23, *p* = .819; and dietary quality, *t*(958) = −0.30, *p* = .767. 

Second, as food neophilia was included as a latent variable in the following cross-lagged panel analysis, confirmatory factor analyses (CFA) were conducted to test the measurement invariance (MI) of the latent construct of food neophilia across both waves, ensuring valid and meaningful across-time comparisons [58]. The MI was tested at the levels of configural, metric, scalar, and residual invariance by using a multi-step approach comparing a set of nested latent structural equation models, using all of the VARSEEK items as indicators for the latent construct. First, assuming a configural MI, we specified an unrestricted baseline model with autocorrelated errors among repeatedly measured indicators to account for the variance that an indicator shared with itself over time [59]. In the following, we compared several increasingly constrained models in which the factor loading, intercepts, and measurement error variances of the configural model were gradually set to be equal over time (thus representing metric, scalar, and residual MI, respectively). The evaluation of the model fit was based on multiple indices: the comparative fit index (CFI), the root mean square error of approximation (RMSEA), and the standardized root mean square residual (SRMR). A good model fit is indicated by a RMSEA coefficient of less than .06, a CFI above .95, and a SRMR of less than.08 [60]. As χ^2^ statistics are highly sensitive to large sample sizes, the model comparisons were based on absolute differences in the fit indices. Following Chen [61], a change of ≥−.010 in the CFI, supplemented by a change of ≥.015 in the RMSEA, or a change of ≥.030 in the SRMR for weak MI (and ≥.010 for strong and strict MI) indicates non-invariance. As shown in Appendix A, all of the models yielded a good fit to the data. Moreover, the model comparisons showed that restricting the models did not worsen the model fit, supporting residual invariance over time, and thus allowing for meaningful comparisons of food neophilia over time. The following analyses are based on the residual model.

Third, a cross-lagged panel analysis was conducted to analyze both the temporal stability of food neophilia and dietary quality, as well as the reciprocal effects between the two constructs, using structural equation modeling (SEM). Whereas food neophilia was included as a latent variable in the model, the NutriAct diet score was included as a manifest variable to measure the dietary quality. Within the model, autoregressive and cross-lagged paths were estimated, allowing for an evaluation of the reciprocal effects between food neophilia and dietary quality over time while simultaneously controlling for the stability of the two constructs [58]. We included the participants’ gender, age, body mass index (BMI), educational status, and stringency of government containment policies at T2 as covariates in the model to statistically control for these variables. Due to the wide range of BMI values observed among the participants (see Table 1), supplementary exploratory multigroup comparisons were carried out to test for differences between the different weight status groups. For this purpose, the weight status at T1 was divided into three BMI categories (underweight: BMI < 18.5 kg/m^2^, average weight: BMI = 18.5–24.9 kg/m^2^, overweight and obesity: BMI > 25.0 kg/m^2^). However, due to the small sample size of participants who were underweight (*n* = 13), these cases were excluded from our analysis. Post-hoc analyses using Wald tests were applied to test for significant differences of the path coefficients.

As the participants were nested within families, we accounted for the potential non-independence of the observations by using the robust maximum likelihood estimator (MLR) in combination with a type “complex” option in Mplus (specifying family ID as a cluster variable) for all of the analyses. This approach provides adjusted standard errors and test statistics that are robust to non-normality and non-independence of observations [62]. All of the analyses were performed using Mplus 7 [63]. All of the tests adopted a significance level of .05.

### 2.5. Missing Data

All of the participants who participated in the T1 wave were included in the analyses. To account for missing data, we used a multiple imputation approach to impute the missing data at both time points, including the missing values at T1 and missing data from participants who dropped out of the study. We had the complete data for the NutriAct diet score at both T1 and T2, as answering the nutrition-related questions was a prerequisite for the further processing of the questionnaire. Missing values for the VARSEEK score were very low (99.4% and 99.2% complete data sets for T1 and T2, respectively). These data were missing completely at random, as determined by Little’s [64] Missing Completely at Random (MCAR) test: χ2(41) = 35.635, *p* = .707 for T1 and χ2(28) = 16.422, *p* = .959 for T2. 

Following Geiser [58], 50 imputed data sets were generated, which allowed us to perform robust analyses. All of the subsequent analyses, including the descriptive analyses, measurement invariance analysis, cross-lagged panel analysis, and exploratory multigroup comparisons were based on these imputed data sets. Multiple imputation is a regression-based technique that is considered to be a state-of-the-art method for dealing with missing data [65], and outperforms traditional missing data techniques, such as listwise or pairwise deletion and single imputation techniques [66]. 

## 3. Results

### 3.1. Descriptive Analyses

The bivariate correlations of the study variables food neophilia and dietary quality at T1 and T2, as well as their descriptive statistics (means and standard deviations) and ICCs, are presented in Table 2. Our analyses showed that all of the variables were significantly correlated. The food neophilia at T1 and T2 were each slightly positively associated with the dietary quality at T1 and T2. In addition, there were strong positive correlations between food neophilia at T1 and T2, as well as between dietary quality at T1 and T2.

The analyses showed a significant difference between the T1 and T2 means of food neophilia, Wald(1) = 23.13, *p* < .001, *d* = 0.31, but no significant difference between the T1 and T2 means of dietary quality, Wald(1) = 0.39, *p* = .531. In other words, the participants’ food neophilia slightly decreased on average from T1 to T2, whereas there was no significant change in the mean dietary quality from T1 to T2. The ICCs, as a quantitative measure of the similarity among observations within families, were low for both food neophilia and dietary quality. However, the ICCs indicate a non-independence of observations, which was statistically accounted for in all of the further analyses (see Section 2.4 for more details).

### 3.2. Temporal Stability and Reciprocal Effects of Food Neophilia and Dietary Quality

The cross-lagged panel model of food neophilia and dietary quality is shown in Figure 1, with the standardized parameters reported. We additionally controlled for the participants’ gender, age, BMI, educational status, and stringency of government containment policies at T2 by including these variables as covariates in the model (they are omitted from Figure 1 for clarity). The model yielded a good fit to the data: CFI = .953, RMSEA = .054, SRMR = .039. To test for robustness, we additionally estimated the model using a full-information maximum likelihood (FIML) estimation to handle the missing data. This model fully replicated the critical pathways determined with the multiple imputation procedure (CFI = .956, RMSEA = .049, SRMR = .039). Supporting our hypotheses, the food neophilia at T1 predicted the food neophilia at T2, and the dietary quality at T1 predicted the dietary quality at T2, indicating high interindividual stability of both constructs. In addition, the analyses revealed a small positive cross-sectional correlation between food neophilia and dietary quality. Controlling for the temporal stability and the cross-sectional correlation, food neophilia did not have a significant prospective effect on dietary quality. Dietary quality showed a very small positive prospective effect on food neophilia. The model explained 67.0% of variance in food neophilia and 44.5% of variance in dietary quality.

### 3.3. Exploratory Multigroup Comparisons

The model of multigroup comparisons showed an acceptable fit to the data (CFI = .953, RMSEA = .056, SRMR = .044). As shown in Figure 2, the results were comparable to those of the total sample. However, our results revealed a positive cross-sectional correlation between food neophilia and dietary quality only in individuals with overweight and obesity, while this association was not significant in individuals with normal weight. Additionally, a small positive prospective effect of dietary quality on food neophilia was observed only in individuals with overweight and obesity. Further post-hoc analyses using Wald tests showed that the only significant difference between the two weight status groups was in the cross-sectional correlation between food neophilia and dietary quality, Wald (1) = −0.26, *p* = .002. 

## 4. Discussion

The present two-wave longitudinal study was designed to explore both the temporal stability of food neophilia and dietary quality, as well as their prospective reciprocal relationship, in a large sample of older adults over a three year period. Our analyses showed a high longitudinal stability of food neophilia and dietary quality and a positive cross-sectional relationship between the two constructs. Food neophilia had no prospective effect on dietary quality. However, we found a very small positive prospective effect of dietary quality on food neophilia. In summary, our findings allowed a deeper understanding of food neophilia and dietary quality in later life and shed light on their interplay over time.

Given the paucity of studies in this field of research, our results offered initial insights into food neophilia and dietary quality in older age. The mean scores of food neophilia indicated a moderate willingness of the participants to try unfamiliar foods at both time points and were slightly lower compared to recent findings on food neophilia in a German community sample aged 18 years and older [35]. Lower scores on food neophilia could be due to the older age or other characteristics of our sample. Our analyses revealed that food neophilia slightly decreased on average over the three year study period, indicating that participants became slightly less willing to try unfamiliar foods from one time point to another. With regard to the related construct of food neophobia, the lifespan model postulated by Dovey et al. [67] assumes that the aversion to eat novel foods reaches a stable level in early adulthood before increasing again in later adulthood, although empirical longitudinal data to support this assumption are lacking so far. One possible explanation for our results may be that health-related concerns increase with age [68], possibly leading to a decreased overall willingness to try unfamiliar foods that may carry the risk of affecting one’s well-being (e.g., gastric distress). Another interesting aspect of the stability of food neophilia in older age can be inferred from the results of our cross-lagged panel analysis, which indicated a very high longitudinal stability, i.e., little change in the interindividual differences across time [69]. In other words, despite the slight decrease in the participants’ average food neophilia over time, each participant’s relative standing on the construct has changed very little, i.e., those with a high food neophilic tendency remained highly food neophilic, and those with a low food neophilic tendency remained less food neophilic. It is conceivable that, particularly in this age group, long-standing and established everyday routines result in little change in food neophilic tendencies and behaviors in daily life. In terms of potential intervention opportunities, one approach to promoting food neophilia in older age may be to bring change to these daily routines. This could be achieved, for example, by deliberately creating everyday situations that involve exposure to novel foods and cuisines (e.g., when grocery shopping or visiting restaurants). Other potential ways to increase food neophilia could include increasing food familiarity, nutrition knowledge, and cooking skills, as well as providing information (especially to older adults) about the potential health benefits of novel foods [70]. More research is needed to test our assumptions and deepen our understanding of potential intervention strategies.

Regarding the examination of the temporal stability of dietary quality in later life, our analyses showed no change in the mean dietary quality over the three year study period, with the participants being moderately adherent to a healthy diet at both time points, as measured by several key dietary determinants of chronic disease risk [19]. In addition, the results of our cross-lagged panel analysis indicated a high longitudinal stability of dietary quality, i.e., little change in the interindividual differences across time. Our results add to the limited research on the temporal stability of dietary quality in later adulthood [15,18] and provide a useful basis for further research. Although our analyses showed no deterioration in dietary quality with age, the moderate dietary quality of the participants on average underscores that older adults are an important target group for promoting health-beneficial eating, particularly in light of the increasing risk of chronic diseases and multimorbidity with age [6]. While the analyses showed a high degree of stability of interindividual differences over time, they do not allow conclusions to be drawn in regard to the individual trajectories of change. Future prospective studies may contribute to a deeper understanding of the potential for favorable dietary change in older age [8] by focusing on a more in-depth analysis of the unique trajectories of change in individuals (or groups of individuals), as well as potential predictors using other appropriate statistical approaches, such as latent growth curve (LGC) analysis [58]. 

In addition to the temporal stability of food neophilia and dietary quality, a special focus of the present work was the reciprocal relationship between the two constructs. In line with the limited previous evidence on their cross-sectional association [39,40], our preliminary analyses showed a small positive correlation between food neophilia and dietary quality at both time points. Interestingly, our exploratory multigroup comparisons revealed that the positive cross-sectional association between food neophilia and dietary quality was evident only in individuals with overweight and obesity, but not in individuals with normal weight. These results suggest that while a high dietary quality appears to be associated with an increased willingness to try new and unfamiliar foods in individuals with overweight and obesity, food neophilia does not seem to play an important role in promoting healthy eating behavior in individuals with normal weight. Our findings may have implications for future research exploring the interplay between food neophilia and dietary quality, particularly with regard to how this relationship may differ based on weight status. One possible avenue for further investigation could be to explore whether interventions aimed at increasing food neophilia to promote healthy eating are more effective in individuals with overweight and obesity than in those with normal weight. In addition, including a larger sample of individuals with underweight in future studies may prove useful to provide a more comprehensive understanding of how the interplay between food neophilia and dietary quality operates across the entire weight range. 

Statistically controlling for the temporal stability of the constructs, as well as their cross-sectional intercorrelation, our analyses did not identify food neophilia as a significant determinant of dietary quality over time. In other words, food neophilia did not prospectively predict dietary quality in our cross-lagged panel analysis. As research on this topic is very scarce, our analyses were conducted in an exploratory manner, and we can only speculate on the reasons why we did not find a prospective effect of food neophilia on dietary quality. 

One possible explanation is that food neophilia per se may indeed not have a major positive long-term impact on overall dietary quality. Food neophilia describes the overt willingness to try unfamiliar foods combined with a general interest in new cuisines, recipes, and dishes [35]. However, no distinction is made as to whether these unfamiliar foods are in fact health-promoting. Given the possibility that unfamiliar foods may also be health-detrimental, it is possible that while willingness to try new foods may be associated with the consumption of a variety of different foods, dishes, and recipes, it may not necessarily contribute to a healthy diet (in the sense of following dietary recommendations to prevent chronic disease). In fact, the internationally accepted recommendation to consume a variety of foods to meet nutrient needs and reduce the risk of nutritional deficiency has increasingly become controversial, as studies on dietary variety have not only shown considerable associations with positive health outcomes [71], but also raised concerns about potential adverse outcomes of diverse diets, such as excess energy intake and obesity [72]. Thus, evidence indicates that dietary variety is not necessarily health-promoting, unless it is embedded in a health-oriented, high-quality diet that includes a variety of foods with a high nutrient density [73,74]. Hence, although we did not find a substantial positive relationship between food neophilia and the overall quality of a diet over time, it may still play an important role in promoting a healthy diet. For example, instead of promoting food neophilia in general, intervention strategies could focus on promoting food neophilia toward nutrient-rich foods specifically, such as unfamiliar fruits and vegetables, as well as healthful food alternatives (e.g., functional foods or nutritionally modified foods). In fact, consumer research indicates that the awareness and acceptance of nutritionally modified and functional products increase with age [75,76], suggesting that older adults are a promising target group for the consumption of healthful food alternatives. Future studies on the relationship between food neophilia and dietary variety within a health-oriented, high-quality diet may help to further elucidate the role of food neophilia in the context of healthy eating. 

Another possible explanation is that there is indeed no considerable prospective reciprocal relationship between the two constructs in later adulthood. Assuming that food neophilia does play an important role in healthy eating, it may have a greater positive impact on dietary quality in the early life stages of childhood, which are considered crucial for the development of health-promoting dietary habits and food preferences [77]. The extensive research on the related construct of food neophobia in children (for an overview see [67]) showed that exposure to novel foods results in an increased acceptance of these foods in everyday diets. Dovey et al. [67] suggested that this may not necessarily be the case later in life when dietary habits are already established and manifested. It seems plausible that, in later adulthood, other emerging factors, such as health cognitions and food-related physiology (for an overview see [20]), are more likely to prospectively influence long-standing dietary habits, and thus dietary quality. Nevertheless, it is possible that promoting food neophilia may contribute to the development of health-promoting dietary habits, not only in childhood, but also at later stages of life. As for the age group of older adults, research suggests that certain changes in an individual’s later life, such as the retirement transition [8,12], marital transitions [9,10], or changing health conditions [11], may provide an effective opportunity to break established (potentially long-standing) unfavorable dietary habits. In contrast to the present study that examined older participants of a relatively broad age range, future prospective studies could specifically target individuals undergoing such life transitions, e.g., those in the peri-retirement age group, to further our understanding of the potential critical windows of opportunity for the promotion of food neophilia. Furthermore, studying participants at multiple time points within a more extended study period could provide additional useful information. Such knowledge may prove useful for future studies aimed at investigating targeted intervention measures to promote healthy eating in later life, and thereby reduce the burden of chronic disease and multimorbidity.

The results of our cross-lagged panel analysis suggested a very small positive prospective effect of dietary quality on food neophilia. The fact that the prior level of food neophilia (i.e., its stable portion) is controlled for, allows us to rule out the possibility that the effect was found simply due to the positive cross-sectional relationship between food neophilia and dietary quality [69], suggesting an interrelation (albeit weak) over time. However, the effect size found in this study was very small. Future studies should investigate the longitudinal association between dietary quality and food neophilia in more detail to further our understanding of the underlying developmental processes. These investigations may also include potential mediating variables (e.g., cooking and food skills [39], nutrition knowledge [41]).

### Strengths and Limitations

The present study was the first to investigate the stability of food neophilia and dietary quality in older adults and their cross-lagged relationship over time. The results of our study should be interpreted in the context of its strengths and limitations. One of the key strengths of our study was its comprehensive dietary assessment strategy, which combined both short-term and long-term dietary assessment tools. Its evaluation system was not only based on the recommendations of the German Nutrition Society, but also included the current evidence for chronic disease prevention, making a distinct contribution to the few previous studies on food neophilia in the context of nutritional research, which have mainly used brief screening tools to measure dietary quality. Although the newly developed NutriAct diet score should be evaluated for its health benefits in future prospective studies, it already represents a valuable advancement in the measurement of adherence to a health-promoting diet in Germany. Another strength was the fact that the present study followed an interdisciplinary approach, combining both psychological and nutritional expertise. In addition, it included two data waves, covering a time span of over three years, as well as a large sample of almost 1000 older adults. Moreover, the number of dropouts in our study was low (<14%) and no systematic dropout bias was evident. Missing values were handled by the state-of-the-art multiple imputation procedure, resulting in less biased parameter estimates compared to traditional missing data techniques. In addition, the application of a cross-lagged panel analysis enabled the bidirectional analysis of food neophilia and dietary quality while controlling for their temporal stability and cross-sectional association. 

Some limitations need to be acknowledged. The first limitation is the use of self-reported data, which can lead to systematic bias [78]. However, the design and concept for this study were carefully planned [44] to minimize information bias [79]. For example, short-term and long-term dietary assessment techniques were combined to assess dietary intake, as this was shown to yield less biased estimates than stand-alone instruments [80]. Moreover, food neophilia was assessed using the psychometrically validated German version of the VARSEEK [35]. In addition, we performed latent modeling of the construct of food neophilia, which allowed us to account for measurement error and to verify its temporal measurement invariance, enabling meaningful comparisons of food neophilia over time. Another limitation is that the data collection for the second wave of the NFS took place between September 2020 and November 2021, i.e., during the COVID-19 pandemic, imposing a whole new set of challenges, including the direct implications on one’s lifestyle, and thus on nutrition and eating behavior [81]. However, a recent systematic review suggests rather inconsistent results regarding the effect of the COVID-19 pandemic on dietary patterns and eating behavior in non-clinical samples [82]. Despite the lack of studies on food neophilia during the pandemic, given the high temporal stability of food neophilia, it is conceivable that the lockdowns proposed by governments during this time negatively impacted the opportunities to consume unfamiliar foods and dishes, but had less of an impact on the actual willingness to try them and its effect on healthy eating. Nevertheless, to statistically control for the potential effects of the pandemic, we added the stringency of government containment and closure policies in Germany during the second data wave as a covariate in our analyses, using the OxCGRT stringency index [53]. A further limitation concerns the representativeness of the sample, with participants showing a high level of education compared to the German population of adults over 50 years [83]. As previous studies have found a positive relationship between educational level and both food neophilia [42] and dietary quality [84], our results may thereby not be generalizable to individuals of all educational levels. Therefore, to increase the generalizability of our results, future studies should include more heterogeneous samples.

## 5. Conclusions

The current work was the first to examine the stability of food neophilia and dietary quality in older, age as well as their reciprocal relationship over time, in a cross-lagged panel design. Our analyses revealed a high longitudinal stability of both food neophilia and dietary quality over a three year period. In addition, we found a small positive cross-sectional correlation between food neophilia and dietary quality, suggesting that food neophilia may play a role (albeit a small one) in the context of a health-promoting diet in older age. However, our post-hoc analyses revealed that this association was evident only in individuals with overweight and obesity, but not in individuals with normal weight. The analyses indicated that food neophilia had no prospective effect on dietary quality, whereas we found a very small positive prospective effect of dietary quality on food neophilia. In summary, the present study allowed a deeper understanding of food neophilia and dietary quality in older age and gave initial insights into their interrelation over time. As our analyses did not identify food neophilia as a significant determinant of dietary quality over time in older age, it will prove useful to understand the developmental trajectories of both constructs in more depth before investigating the potential of food neophilia for intervention strategies to promote healthy eating. In addition, future studies should expand our findings by further researching the relationship between food neophilia and dietary variety within a health-oriented, high-quality diet, as well as the potential critical windows of opportunity for the promotion of food neophilia (e.g., during critical life transitions). Additionally, it would be valuable to explore the mediating role of weight status in the interplay between food neophilia and dietary.

## Figures and Tables

**Figure 1 nutrients-15-01248-f001:**
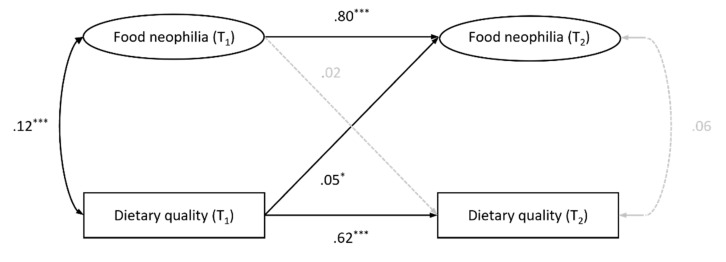
Temporal stability and reciprocal effects of food neophilia and dietary quality (*N* = 960). Standardized path coefficients are reported, controlled for gender, age, BMI, educational status, and stringency of government containment policies at T2. Dotted, gray lines indicate non-significant results (*p* > .05). * *p* < .05. *** *p* < .001.

**Figure 2 nutrients-15-01248-f002:**
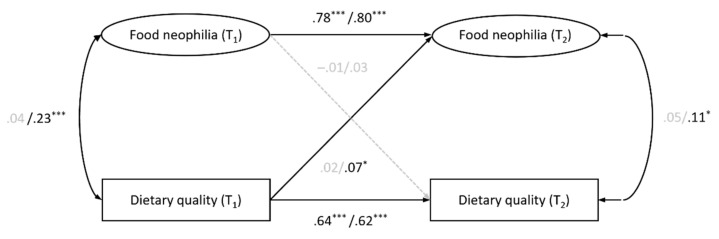
Temporal stability and reciprocal effects of food neophilia and dietary quality for individuals with normal weight (*n* = 429, named first) and with overweight/obesity (*n* = 518, named second) ^a^. Standardized path coefficients are reported, controlled for gender, age, educational status, and stringency of government containment policies at T2. Dotted, gray lines indicate non-significant results (*p* > .05). * *p* < .05. *** *p* < .001. ^a^ Due to the small sample size of participants with underweight (*n* = 13), these cases were excluded from our analysis.

**Table 1 nutrients-15-01248-t001:** Sample characteristics of the NFS at T1 and T2.

		Time 1 (T1)	Time 2 (T2)
Total sample (*N*)		960	829
Gender (*n* (%))			
	Women	512 (53.3%)	442 (53.3%)
	Men	447 (46.6%)	386 (46.6%)
	Nonbinary	1 (0.1%)	1 (0.1%)
Age (years)			
	*M* (*SD*)	63.4 (6.1)	66.9 (5.9)
	Min–Max	50–84	53–88
BMI (kg/m^2^) ^1^			
	*M* (*SD*)	25.96 (4.13)	25.79 (4.18)
	Min–Max	15.09–46.30	15.18–58.81
Educational status ^2^ (*n* (%))			
	Low	31 (3.2%)	25 (3.0%)
	Medium	297 (30.9%)	251 (30.3%)
	High	632 (65.9%)	553 (66.7%)

^1^ BMI was calculated from self-reported height and weight (BMI = weight (kg)/height (m)^2^); ^2^ Educational status was measured by the three-level CASMIN-index [45].

**Table 2 nutrients-15-01248-t002:** Intercorrelations, descriptive statistics, and ICCs of manifest food neophilia and dietary quality at T1 and T2 (*N* = 960).

		1.	2.	3.	4.
1.	T1 food neophilia	1			
2.	T1 dietary quality	.12 ***	1		
3.	T2 food neophilia	.80 ***	.14 ***	1	
4.	T2 dietary quality	.07 *	.64 ***	.10 **	1
*M* (*SD*)		4.30 (1.41)	5.59 (0.88)	4.15 (1.36)	5.60 (0.95)
ICC		.21 ***	.31 ***	.23 ***	.26 ***

* *p* < .05. ** *p* < .01. *** *p* < .001.

## Data Availability

As this study is part of the ongoing NFS, public access to the data presented in this study will be arranged upon reasonable request and with the permission of all collaboration partners.

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
