# Peer review of "Exploring the Longitudinal Stability of Food Neophilia and Dietary Quality and Their Prospective Relationship in Older Adults: A Cross-Lagged Panel Analysis"

_nutrients, 2023, doi:10.3390/nu15051248_

Round 1

Reviewer 1 Report

Thank you for allowing me to review this manuscript titled Exploring the longitudinal stability of food neophilia and dietary quality and their prospective relationship in older adults: A cross-lagged panel analysis. 

The topic is relevant and fits the journal scope. The authors conducted an interesting study with results that can fill a gap in the literature. The manuscript is well written. I have some minor comments to make the manuscript more readable. 

Introduction

This section has all the elements to understand the main study variables and the importance of studying this topic. But, sometimes, the introduction feels long with repeated ideas. I suggest reducing some ideas in the text. 

Material and methods

In table 1, the range indicated is the minimum and maximum, not the range. 

Results

From the methods section, I understood that the total sample the authors worked with was 829 (those participants with two times measurements). However, in all tables, it appears a sample size of 960. Please, clarify. 

Also, I suggest moving sections 3.2 and 3.3 to the methods section. Also, I recommend summarizing these sections information. Some of this could be an annex. 

Lines 300 and 301. How did the authors consider the non-independence of observation in the results? They indicate to review section 2.4 for details, but this is not explained there either. 

Table 2. The significance level is not shown in the M(SD) values. 

Discussion

The authors state several limitations. However, they noted how they overcame some of them. If they overcame limitations, these were not limitations anymore. 

Reviewer 2 Report

In the current study the authors intented to explore the prospective relationship between food neophilia and dietary quality  in a group of German adults aged 50-84 years old.  The results showed that food neophilia remained relatively stable over a period of ~3+ years, and positive correlations between dietary quality and  food neophilia were observed cross-secionally, yet food neophilia was not a predictor of dietary quality prospectively.  The significance of the current study is to provide some evidence that there may be reciprocal relationships between dietary quality and food neophilia in the older adults, which may open a window for a new direction for the developing strategies to improve dietary quality in the elderly.

1.  The authors stated that the sample sizes are 960 from 409 families at T1, and 829 from 372 families at T2, respectively (lines 172-173).  Is it possible to report the number of participants from the same families in more detail (for example, how many families with two, three or even more participants)?

2. Socialeconomic status (SES) has been considered an important factor influencing dietary quality (and perhaps the tendency of food neophilia as well).  In addition to the educational status, is there any other SES index, such as annual income and/or housing status (or a combination one) available?

3. As the participants included in the current study aged 50-84 years, there should be at least part of them still working; these relatively "younger" participants may differ from those who already retired in the aspects of social connection, sources of obtaining health information, and so on.  The authors may want to try to analyze whether the relationships between dietary quality and food neophilia differ in the subgroups.

4. The participants included in the study have a wide range of BMI (Table 1).  Though the authors have controlled for the effect of BMI in their analyses, it may still be insteresting to perform additional analyses on those with abnormal BMIs, especially those who were underweight.

5. In general, the authors found that a small but positive prospective effect of dietary quality on food neophilia, yet no prospective effect of food neophilia on dietary quality on the other hand.  Is it possible to perform some further analses in the subgroup of participants with lower dietary quality?  As this may be a group with higher risk of poorer nutrition status and may need more effective strategies to intervene.  

Reviewer 3 Report

First of all I would like to thank you for the opportunity to review this research. I think this research is very interesting and fits the theme of this journal.

In line with the theoretical framework, the variables are very well adapted and contextualised.

As an area for improvement, I would add the research hypotheses established at an early stage. 

With regard to the methodological contextualisation, this is carried out correctly and meets all the criteria of the scientific method. As aspects of improvement for future interventions, I would use a Bioimpedance scale to calculate BMI. Add the programme used to carry out the statistical analyses and the structural equation model. 

Finally, I recommend replacing all pre-2005 references with more current research. 
